# Comparative Study on CFD Turbulence Models for the Flow Field in Air Cooled Radiator

**Chao Yu** *[ID], **Xiangyao Xue, Kui Shi, Mingzhen Shao and Yang Liu**

Changchun Institute of Optics, Fine Mechanics and Physics, Chinese Academy of Sciences, Changchun 130033, China; my691930402@163.com (X.X.); qw417345721@163.com (K.S.); ou910160194@163.com (M.S.); fg576856647@163.com (Y.L.)
* Correspondence: yuchao@ciomp.ac.cn

**Abstract:** This paper compares the performances of three Computational Fluid Dynamics (CFD) turbulence models, Reynolds Average Navier-Stokes (RANS), Detached Eddy Simulation (DES), and Large Eddy Simulation (LES), for simulating the flow field of a wheel loader engine compartment. The distributions of pressure fields, velocity fields, and vortex structures in a hybrid-grided engine compartment model are analyzed. The result reveals that the LES and DES can capture the detachment and breakage of the trailing edge more abundantly and meticulously than RANS. Additionally, by comparing the relevant calculation time, the feasibility of the DES model is proved to simulate the three-dimensional unsteady flow of engine compartment efficiently and accurately. This paper aims to provide a guiding idea for simulating the transient flow field in the engine compartment, which could serve as a theoretical basis for optimizing and improving the layout of the components of the engine compartment.

**Keywords:** computational fluid dynamics; heat transfer; engine compartment; vortex; turbulent model

## 1. Introduction

The turbulent structure is very important to the flow field distribution of the power compartment of the loader. Its transportation velocity and distribution have important research values. The small-scale turbulent structure is transported at the speed of local flow field, while the large-scale turbulent structure maintains a fixed proportion between the transport speed and the average flow velocity [1].

Bull et al. [2] conducted experiments on subsonic boundary layers, and the results showed that the proportion between the transport velocity and the average flow velocity of the small-scale turbulent structure was 0.53, while that of large-scale turbulent structure was 0.825. Favre et al. [3–5] have extensively studied the spatio-temporal correlation characteristics of boundary layer turbulent fields. The results show that the high and intermediate frequency pulsations contain about 50% turbulent kinetic energy, which is mainly characterized by high viscous dissipation and strong space-time attenuation, while the low frequency pulsations contain another 50% turbulent kinetic energy, of which only 1% is lost due to viscosity. Demetriades [6] conducted an experiment on the compressible supersonic boundary layer, which shows that the spatial correlation between flow direction and radial direction is basically the same, and only after the distance of vortex structure migration exceeds its own scale does the correlation significantly decrease. Ganapathisubramani et al. [7] measured the boundary layer with particle image velocimetry (PIV), and the results showed that the flow direction and the spreading scale in the boundary layer increased with the increase of the distance to the wall, and the change rate was basically linear. In the external flow field, the trend of scale variation is the opposite. The lower speed zone contributes more to the attenuation of flow correlation Ruu. Kline et al. [8] observed that there were a large number of low-velocity strip structures in the boundary layer through

the $H_2$ bubble experiment. Falco et al. [9] studied the quasisturon motion in the turbulent boundary layer based on the smoke spectrum, determined the motion scale, and observed a large number of vortex structures. Moin and Kim [10] carried out a simulation study on turbulent channel flow based on large eddy simulation. Thereafter, numerical studies based on DNS were carried out in large quantities [11–13].

Different from traditional experimental method, numerical technology for the data of the turbulent flow field are measured and stored. The data processing means a great deal of diversity, and it can measure all direct physical quantities, such as pressure, velocity, and deals with predefined indirect quantities. For example, in the previous experiment, single point measurement was carried out based on thermocouple anemometer, so quadrant analysis could only be limited to a few measurement points, and it was concluded that up-jet and down-sweep were upstream and downstream relationships [14]. Numerical studies will not be subject to such restrictions, and not only can we obtain the three-dimensional structure of the quadrant component of specific instantaneous shear product, we can also determine the vortex structure [15–17]. Compared with the experiment, the biggest advantage of numerical study lies in the traceability of specific experimental phenomena, that is, it can trace forward and deduce the quasi-structure and quasi-motion at a certain time point. At the same time, it can also be studied by the means of defining phenomena and using conditional sampling [18]. Calautit J. K. et al. [19] used virtual wind tunnels to replace traditional wind tunnel tests to, respectively, conduct flow field and transmission of the power cabin of passenger cars and loaders. Lu P. et al. [20] adopted the method of one-dimensional and three-dimensional co-simulation to analyze the flow characteristics of the power cabin and the change rules of the internal thermal environment.

However, different turbulence models are used to predict uniformity, which results in different results. Therefore, hybrid meshing inside the engine compartment is used in this research, and a lot of energy and time is spent to improve the grid quality. By comparing it with the experimental results, the prediction performance of different turbulence models on the flow field and temperature field of the engine compartment is evaluated. The temperature field, pressure streamline structure, vorticity, and velocity distribution in the engine compartment are analyzed. Compared with the Reynolds-averaged Navier-Stokes equations (RANS) model, the Large Eddy Simulation (LES) and Detached Eddy Simulation (DES) models can capture transient vorticity characteristics such as the generation, development, and formation of vortices, as well as the trailing edge shedding and fracture. In addition, the DES model has been proven to be a feasible method to accurately and efficiently simulate three-dimensional unsteady flow in complex cabin space.

## 2. Numerical Simulation

### 2.1. Turbulence Models

The average time domain of the Navier-Stokes equation to obtain the continuity and momentum equations for incompressible flow is as follows:

$$\frac{\partial \rho}{\partial t} + \frac{\partial(\rho u_i)}{\partial x_i} = 0$$
$$\frac{\partial(\rho u_i)}{\partial t} + \frac{\partial(\rho u_i u_j)}{\partial x_j} = -\frac{\partial p}{\partial x_i} + \frac{\partial}{\partial x_j}\left(\mu \frac{\partial u_i}{\partial x_j}\right) + \frac{\partial \tau_{ij}}{\partial x_j} \tag{1}$$

where $\rho$ denotes the density and $u_i$ denotes velocity. $p$ is the pressure, and $\mu$ is the coefficient of viscosity.

Where the Reynolds stress $\tau_{ij}$ is:

$$\tau_{ij} = -\rho \overline{u_i u_j}. \tag{2}$$

The most widely used turbulence model is the standard $k$-$\varepsilon$ model. The model is based on the kinetic energy $k$ equation, and then an equation of turbulent dissipation rate $\varepsilon$ is introduced. The $\varepsilon$ is defined as:

$$k = \frac{1}{2}mv^2, \tag{3}$$

$$\varepsilon = \frac{\mu}{\rho}\overline{\left(\frac{\partial u'_l}{\partial x_k}\right)\left(\frac{\partial u'_l}{\partial x_k}\right)}. \tag{4}$$

The turbulent viscosity $\mu_t$ can be expressed as a function of $k$ and $\varepsilon$:

$$\mu_t = \rho C_\mu \frac{k^2}{\varepsilon}. \tag{5}$$

Therefore, the transport equation of the standard $k$-$\varepsilon$ model is [21]:

$$\begin{cases} \frac{\partial(\rho k)}{\partial t} + \frac{\partial(\rho k u_i)}{\partial x_i} = \frac{\partial}{\partial x_j}\left[\left(\mu + \frac{\mu_t}{\sigma_k}\right)\frac{\partial k}{\partial x_j}\right] + G_k + G_b - \rho\varepsilon - Y_M + S_k \\ \frac{\partial(\rho\varepsilon)}{\partial t} + \frac{\partial(\rho k u_i)}{\partial x_i} = \frac{\partial}{\partial x_j}\left[\left(\mu + \frac{\mu_t}{\sigma_\varepsilon}\right)\frac{\partial\varepsilon}{\partial x_j}\right] + C_{1\varepsilon}\frac{\varepsilon}{k}(G_k + C_{3\varepsilon}G_b) - C_{2\varepsilon}\rho\frac{\varepsilon^2}{k} + S_\varepsilon \end{cases} \tag{6}$$

where the kinetic energy caused by the average velocity gradient is:

$$G_k = \mu_t\left(\frac{\partial u_i}{\partial x_j} + \frac{\partial u_i}{\partial x_i}\right)\frac{\partial u_i}{\partial x_j}. \tag{7}$$

The turbulent energy generated by the buoyancy effect $G_b$ is [22]:

$$G_b = \beta g_i \frac{u_t}{\mathrm{Pr}_t}\frac{\partial T}{\partial x_i}. \tag{8}$$

The following values are used for standard $k$-$\varepsilon$ model: $C_{1\varepsilon} = 1.44$, $C_{2\varepsilon} = 1.92$, $C_{3\varepsilon} = 0.09$, $\sigma_k = 1.0$, and $\sigma_\varepsilon = 1.3$.

The basic idea of RANS is to perform time averaging on the N-S equations to transform the unsteady turbulence problem into a steady problem study, at the cost of additional unknown numbers, which are also in the same form as stress, called Reynolds stress. Reynolds stress also needs to be characterized by a model, which is the so-called turbulence model. However, due to the time averaging of the problem, the information contained in the equation itself has been partially lost. It is actually very difficult to give a Reynolds stress model and it is also difficult to apply it to all flows.

LES is a commonly used turbulence simulation method. LES is based on the self-similarity theory and uses a subgrid-scale to simulate small vortices, while the large vortices are based on the geometric calculation. The disadvantage of this model is that it is difficult to calculate the region near the wall. In the LES simulation method, the transport of fluid momentum, mass, energy, and other physical quantities are mainly affected by large-scale vortices. Small-scale vortices with isotropic motion are not affected by geometric and boundary conditions. Therefore, LES uses the N-S equation to calculate the turbulent motion larger than the grid-scale. The LES model obtains a filtered momentum equation by filtering out vortices smaller than the filtered grid in the Fourier equation or spatial domain N-S equation. The filtered N-S equation and the continuity equation are transformed into [22]:

$$\begin{cases} \frac{\partial\rho}{\partial t} + \frac{\partial(\rho\overline{u_i})}{\partial x_i} = 0 \\ \frac{\partial}{\partial t}(\rho\overline{u_i}) + \frac{\partial}{\partial x_j}(\rho\overline{u_i u_j}) = -\frac{\partial\overline{p}}{\partial x_i} + \frac{\partial}{\partial x_j}\left\{\mu\frac{\partial\overline{u_i}}{\partial x_j}\right\} - \frac{\partial\tau_{ij}}{\partial x_j} \end{cases}. \tag{9}$$

The subgrid-scale stress tensor that results from filtering the Navier-Stokes equations consists of three terms:

$$\tau_{ij} = L_{ij} + R_{ij} + C_{ij}, L_{ij} = \overline{\overline{u_i}\overline{u_j}} - \overline{u_i}\overline{u_j}, C_{ij} = \overline{\overline{u_i}u'_j + u'_i\overline{u_j}}, R_{ij} = \overline{u'_i u'_j} \tag{10}$$

where $\overline{u_i}$ denotes the filtered velocity component and $u'_i = u_i - \overline{u_i}$ denotes the SGS component of $u_i$. $L_{ij}$ is the Leonard term, $C_{ij}$ is the cross term, and $R_{ij}$ is the SGS Reynolds stress.

The sub-grid stress term $\tau_{ij}$ is obtained from the vortex viscous model equation:

$$\tau_{ij} - \frac{1}{3}\tau_{kk}\delta_{ij} = -2\mu_t\overline{S_{ij}} \tag{11}$$

where $\mu_t$ is the turbulent dynamic viscosity, and $\overline{S_{ij}}$ is the rate-of-strain tensor, which is defined as:

$$\overline{S_{ij}} = \frac{1}{2}\left(\frac{\partial \overline{u_i}}{\partial x_j} + \frac{\partial \overline{u_j}}{\partial x_i}\right). \tag{12}$$

Because the turbulence scale decreases rapidly with the increase of Reynolds number, LES simulation cannot be used in engineering calculation at a high Reynolds number. According to the characteristics of the RANS/LES model, a kind of large-scale separated flow in engineering is reasonably arranged by "model allocation." Since the LES model is not an ideal model for solving near-wall problems, the DES turbulence model was developed. The DES turbulence model combines RANS with LES methods. The basic idea of DES is to use the RANS model within the boundary layer and the LES side in the separation area, which saves computational resources and ensures computational accuracy. The DES model selected SST (Shear Stress Transfer) as the RANS model in this study, and its flow energy diffusion term $Y_k$ is modified to [23]:

$$\begin{cases} Y_k = \rho\beta^* kwF_{DES} \\ F_{DES} = \max\left(\frac{L_t}{C_{DES}\Delta_{\max}}, 1\right) \end{cases} \tag{13}$$

where $C_{DES}$ is a calibration constant in the DES model with a value of 0.61, and $\Delta_{\max}$ is the largest grid gap in the three directions ($\Delta x$, $\Delta y$, $\Delta z$) in the grid.

### 2.2. Geometric Model Construction and Mesh Generation

In this paper, different computational fluid dynamics (CFD) turbulence models are used to simulate the flow field of a loader's engine compartment. In order to facilitate the calculation, the model is appropriately simplified, and the components such as the bucket, front axle, and wheels that have less influence on the flow field of the engine compartment are omitted. A 3D simulation model of the engine compartment is established to analyze its internal flow field. The simplified geometric model consists of an outer wind tunnel, an inner wind tunnel, and a loader model as shown in Figures 1 and 2.

The GAMBIT software is used to perform hybrid meshing on the radiator group. Due to the irregular structure of the fan rotation domain and engine compartment, the mesh was built by tetrahedron and hexahedral. The grid of the air intake, inlet fan, and air duct area of the engine compartment are refinement. In order to reduce the total mesh size of the model, a two-layer wind tunnel nested structure is used. The size-Function and Hex-core functions are used to reduce the total mesh number in the outer and inner wind tunnels. The grid distribution in the engine compartment is dense. The outer wind tunnel and the inner wind tunnel mesh are sparse, and the virtual wind tunnel model mesh is shown in Figure 3. The minimum grid size of the simulation model is 1 mm. The total number of grids in the entire calculation model is 800,000.

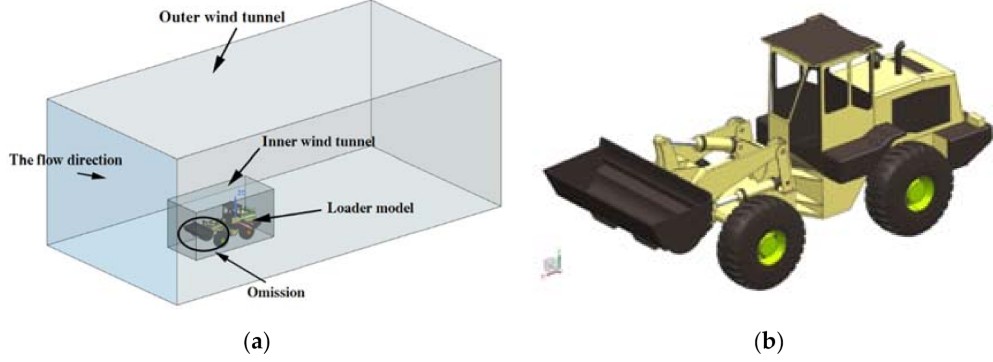

(**a**)                           (**b**)

**Figure 1.** Virtual wind tunnel model. (**a**) The whole model; (**b**) the wheel loader.

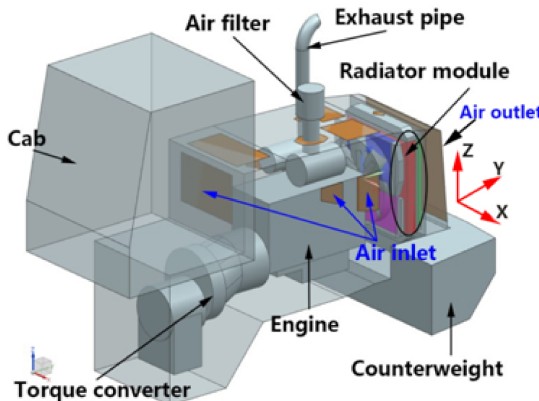

**Figure 2.** Simplified engine compartment.

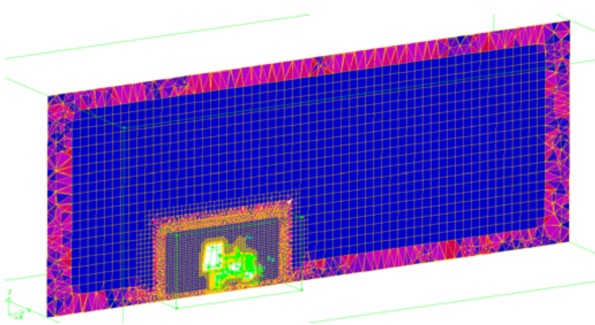

**Figure 3.** Engine compartment virtual wind tunnel model meshing (x = 0 section).

*2.3. Boundary Condition Setting*

The incompressible N-S equation is solved by ANSYS Fluent software. During the calculation, the fan rotates along the mesh section at a certain time step (0.01 s). When the change of mass flow rate in two consecutive circulatory iterations and all normalized residuals in mass and momentum conservation equations were less than $10^{-4}$ for the differences, the solution was assumed to be converged [24–27]. The fan speed is the same as the experimental value. The mesh interface is set between the various modules of the engine compartment. The boundary conditions and settings of the CFD solver are shown in Table 1. The radiators in the engine compartment are simulated by a porous medium model [28].

**Table 1.** CFD (computational fluid dynamics) model description.

| Analysis Type | Transient State |
| --- | --- |
| Turbulence Model | LES, DES, and Standard $k - \varepsilon$ model |
| Pressure-Velocity Coupling | SIMPLEC |
| Transient Formulation | Second-order implicit |
| Fan Status | Varied from 0 rpm to 2000 rpm |
| Vent Status | Outflow |
| Environment Temperature | 25 °C |
| Inlet velocity | 1.5 m/s |

## 3. Results and Analysis

*3.1. Experiment Analysis*

The credibility of the computational model is verified by experiments. There are 9 monitoring points set at the exit of the engine compartment. When the fan speed reaches 1000 rpm, they collect the

wind speed value. The monitoring point position is shown in Figure 4. Figure 5 shows the comparison of the three simulation results at 9 monitoring points with the experimental data collected at 1000 rpm. Compared with the experimental results, the error of the calculation results of each simulation model is less than 15%, and the simulation model can solve the actual fluid flow of the loader's engine compartment relatively correctly.

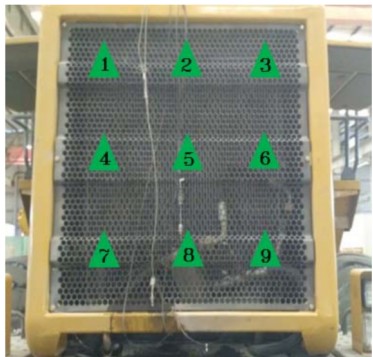

**Figure 4.** Flow rate collection points of experiment.

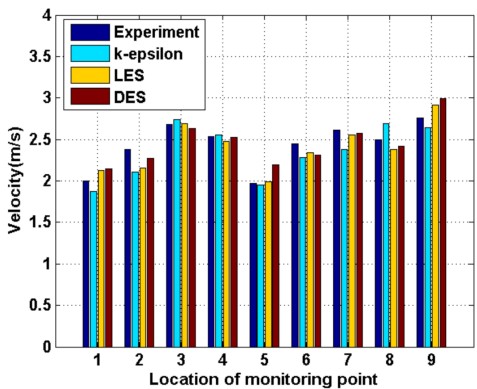

**Figure 5.** Simulation calculation and experimental data comparison (at 2000 rpm).

### 3.2. Pressure Distribution

The arrangement of various heat dissipating components in the loader plays an important role in the pressure distribution of the engine compartment. Figure 6 shows the evolution of pressure in the engine compartment with fan speed. As the fan speed increases, the velocity of the fluid in the cabin gradually increases. A low-pressure region appears at the fan shaft center and a high-pressure region appears at the outer edge of the fan. At low speeds, the surface unevenness of the radiator is consistent with the dynamic pressure of the fan outlet. Due to the large space in the upper part of the engine, the fluid inflow is large at high speeds. Under the commonality of the viscous force and pressure gradient, the radiator gradually approaches the equilibrium pressure.

There is a slight difference in the pressure gradient between the RANS, LES, and DES models. Compared with the other two simulation models, the LES model has a significantly low-pressure region at the fan shaft center when fan speeds are high. The pressure trends described in the three turbulence models are the same, indicating that all simulations can effectively capture the mainstream structure.

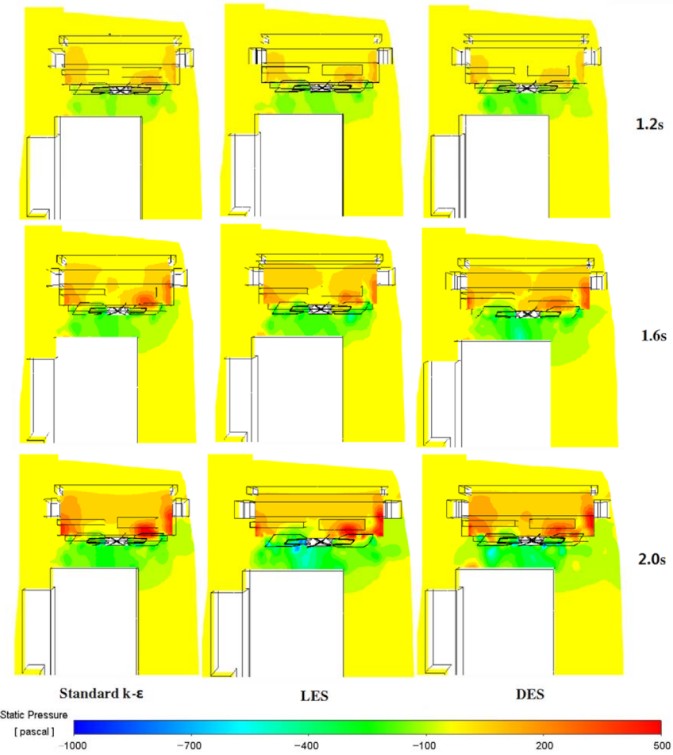

**Figure 6.** 2D contours of pressure magnitude at YZ plane.

### 3.3. Velocity Distribution

The evolution of the fluid flow field in the engine compartment is shown in Figure 7. All simulation models are good at predicting the magnitude and level of flow. However, after a closer look at the YZ section, it can be seen that there is a recirculation zone generated by the intake air in the LES and DES models, while it remains substantially stationary in the RANS model. Figure 8 shows the flow field distribution of the XY section. The RANS model shows multiple small turbulences between the engine and fan, while the LES and DES models are relatively smooth in this area.

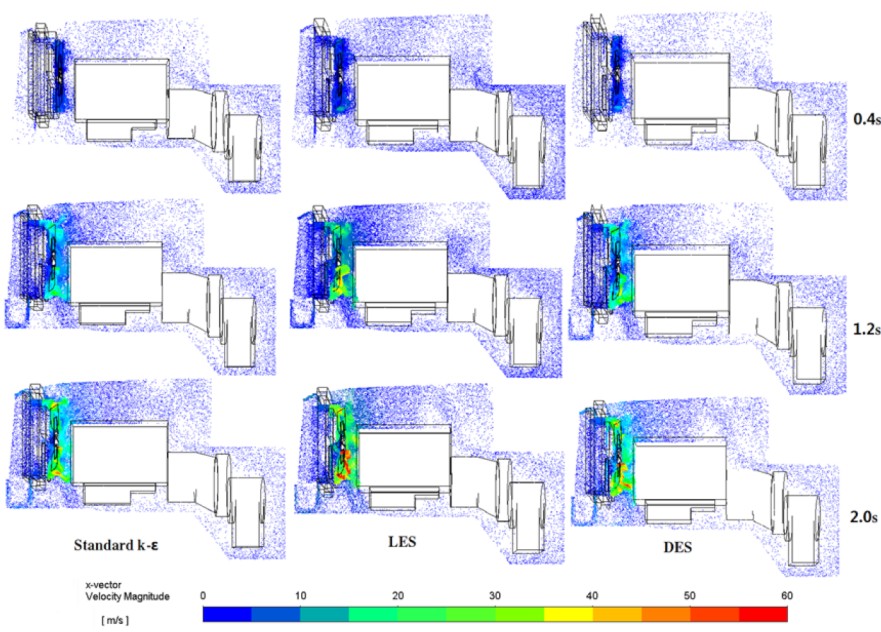

**Figure 7.** 2D contours of velocity magnitude at YZ plane.

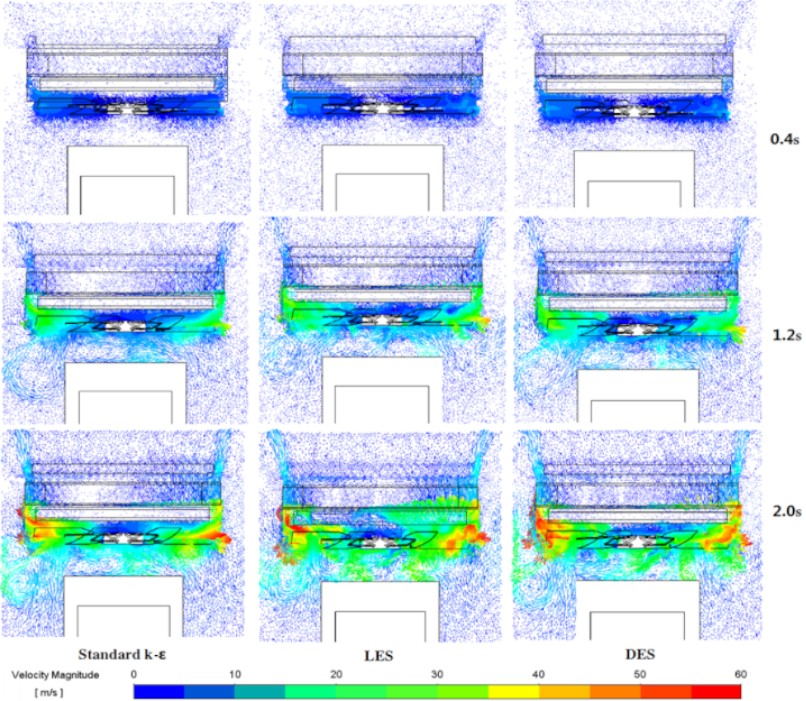

**Figure 8.** 2D contours of velocity magnitude at XY plane.

According to Figures 7 and 8, as the rotational speed increases, the high-speed zone on the outer edge of the fan matches the pressure distribution described above. A high flow rate region occurs at the boundary of the radiator, and the seal between the radiators of each layer allows the cold air to pass sufficiently. There is a retardation zone between adjacent radiators, and the efficient radiator has a relatively small pressure drop retardation zone. In the simulation of the engine compartment, it can be seen that the cold air mainly flows from above and below the power compartment.

*3.4. Vortex Structure*

The vortex was shown by a special isosurface (called vortex core). Based on the Q criterion, the Q criterion index is used to describe it. The recipe is defined as follows [29,30]:

$$Q = \frac{1}{2}(\Omega^2 + S^2) \tag{14}$$

where $\Omega$ and S represent the vorticity tensor and strain rate tensor, respectively.

Figure 9 is a vortex core diagram of the rotating region. As we can see in the standard transient structure for the Q standard, the vortex structure among RANS, LES, and DES models are completely different. LES and DES can reflect more detailed vortex cores.

It can be seen from the figure that the vortex core of the fan is mainly composed of two parts; the vortex core is generated from the blade root vortex and tip vortex formed by the leading edge of the blade. The vortex sheets remain attached on the pressure side while separated at the tip. A high-velocity vortex core appears on the windward surface of the fan. As the rotational speed increases, the fan blade surface gradually forms a vortex. Due to the faster transition between DES and LES, the vortex is also present at the root of the blade, which is gradually developed more abundantly and orderly.

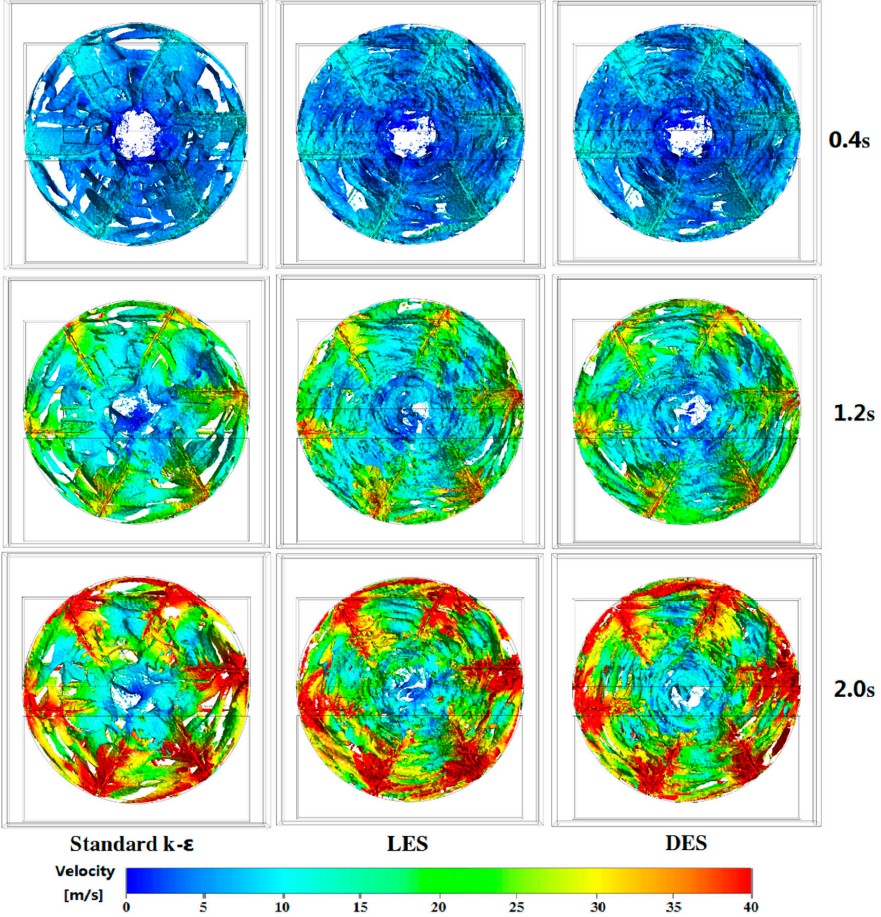

**Figure 9.** Vortex structures of the fan.

### 3.5. Prediction Performance of the Three Turbulent Models

As shown in Figure 10, the simulation data are in good agreement with the experimental data, and the wind speed at each monitoring point increases gradually. A low-speed zone occurs at the detection point 5, which is consistent with the unevenness of the fan flow rate. The prediction speed of the Standard k-ε model and LES model is relatively larger than the DES model. However, the DES model is more accurate than the Standard k-ε model and LES model, whose absolute error is under 20%.

The calculation time of the three simulation models is shown in Table 2. The RANS simulation is about 2.8 times faster than the LES simulation. In industrial design, the RANS model is used for CFD simulation. Although the calculation cost of the RANS model is low, its accuracy is lower than that of the LES model. The LES model has high precision, but the calculation time is long and it is difficult to apply to actual engineering. Besides, the DES model, which combines the advantages of LES and RANS models, is an ideal choice for CFD simulation with short computation time and high accuracy.

**Table 2.** Comparison of computation time with different models (3.4 GHz, 2 CPU).

| Model | CPU Time (h) |
|-------|--------------|
| RANS  | 12.3         |
| DES   | 22.1         |
| LES   | 34.4         |

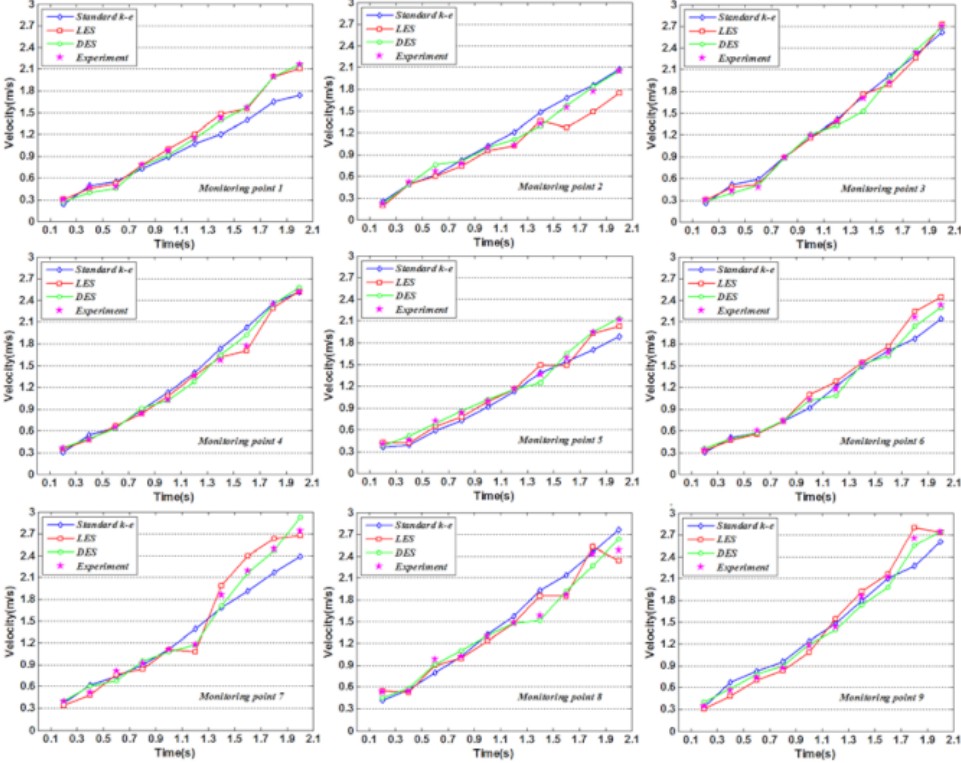

**Figure 10.** Prediction performance of three models.

## 4. Conclusions

In this paper, the hybrid grid model is used to simulate the unsteady flow of the engine power compartment. The ability of the RANS, DES, and LES models to predict the turbulence distribution is compared and analyzed. The engine compartment propulsion simulation using RANS, DES, and LES is proposed. Scholars can use the RANS model to estimate the flow velocity and structure reasonably and accurately. However, LES or DES models should be used when it is necessary to obtain more accurate turbulence distributions with complex geometric structures. Under normal working conditions, the natural wind energy in the environment improves the heat dissipation performance of the engine compartment of the loader. In this study, considering the influence of natural wind on heat dissipation can improve the accuracy of performance prediction. The DES model combines the advantages of the RANS model and LES model, which can accurately simulate a complex eddy current field. At the same time, the LES model can simulate large-scale eddy currents in the boundary layer. The DES model has proven to be an accurate and efficient model for simulating three-dimensional unsteady turbulent flow in complex channels. Meanwhile, theoretical research on this aspect will be the focus of CFD simulation in the future.

**Author Contributions:** Conceptualization, C.Y.; methodology, C.Y.; software, X.X.; validation, K.S.; formal analysis, K.S.; investigation, X.X.; resources, M.S.; data curation, M.S.; writing—original draft preparation, C.Y.; writing—review and editing, C.Y.; visualization, Y.L.; supervision, X.X.; project administration, X.X.; funding acquisition, X.X. All authors have read and agreed to the published version of the manuscript.

**Funding:** This research received no external funding.

**Conflicts of Interest:** The authors declare no conflict of interest.

## Nomenclature

| | | | |
|---|---|---|---|
| $k$ | Turbulent kinetic energy [m$^2$/s$^2$] | $\varepsilon$ | Turbulent dissipation rate [m$^2$/s$^3$] |
| $N$ | Rotating speed [rpm] | $\omega$ | Relative velocity [m/s] |
| $u_i$ | Time-averaged velocity [m/s] | $\nu$ | Turbulent viscosity [m$^2$/s] |
| $X_i$ | Coordinate [m] | $\rho$ | density [kg/m$^3$] |
| $t$ | Time [s] | $\mu$ | Circumferential velocity [m/s] |
| $P$ | Fluid pressure [Pa] | $\mu_t$ | Turbulence viscosity coefficient |
| | | $\tau_{ij}$ | Reynolds stress |

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
