# Peer review of "Comparative Study on CFD Turbulence Models for the Flow Field in Air Cooled Radiator"

_processes, doi:10.3390/pr8121687_

Round 1
Reviewer 1 Report
- Abstract, page 1: the following sentence “The distributions of pressure fields, velocity fields, and vortex structures in a hybrid-grided engine compartment model are analyzed, which reveals that the LES and DES models can capture the detachment and breakage of trailing edge more abundantly and meticulously compared with RANS model.” should be better reformulated.
- Section 2 (page 2): all variables and parameters appearing in Equations (1)-(13) should be defined in the text.
- Section 2 (pages 3-4): how the authors have defined the 5 parameters given at line 95 of page 3 and the one at line 133 of page 4?
- Section 3.4 (page 9): the sentence “between RANS, LES and DES models” should be substituted by “among RANS, LES and DES models”.
Author Response
Dear Reviewer and Editor:
We would like to sincerely thank the reviewer for the very valuable and constructive suggestions. We have revised our manuscript from the suggestions. The following are our responses to the reviewers' comments.
1.Abstract, page 1: the following sentence “The distributions of pressure fields, velocity fields, and vortex structures in a hybrid-grided engine compartment model are analyzed, which reveals that the LES and DES models can capture the detachment and breakage of trailing edge more abundantly and meticulously compared with RANS model.” should be better reformulated.
The sentence has been revised as the reviewer requested and marked red.
2.Section 2 (page 2): all variables and parameters appearing in Equations (1)-(13) should be defined in the text.
All variables and parameters have been revised as the reviewer requested and marked red.
3.Section 2 (pages 3-4): how the authors have defined the 5 parameters given at line 95 of page 3 and the one at line 133 of page 4?
The relevant parameters were selected through references and existing research experience.
4.Section 3.4 (page 9): the sentence “between RANS, LES and DES models” should be substituted by “among RANS, LES and DES models”.
The sentence has been revised as the reviewer requested and marked red.
Reviewer 2 Report
The paper comes to confirm the calculation capabilities of the 3 usedcalculation models. The approach is done gradually following a recipe.
The numerical results are compared at different points with the
experimental ones, in order to validate the calculations, and it is done
successfully. The authors show the advantages and disadvantages of the 3
models applied on a concrete situation.
Although the paper does not bring original elements it still treats the
problem in a correct manner and provides to the engineers and researchers
arguments when choosing to use one or another computational model.
Taking in account the above mentioned elements, I consider that the
paper can be published in its present form.
Author Response
Thanks to the reviewers for approval.